# Sports, Executive Functions and Academic Performance: A Comparison between Martial Arts, Team Sports, and Sedentary Children

**DOI:** 10.3390/ijerph182211745

**Published:** 2021-11-09

**Authors:** Giulia Giordano, Manuel Gómez-López, Marianna Alesi

**Affiliations:** 1Department of Psychology, Educational Science and Human Movement, University of Palermo, 90135 Viale delle Scienze, Italy; marianna.alesi@unipa.it; 2Department of Physical Activity, Sport Faculty of Sports Science, University of Murcia, 30720 Murcia, Spain; mgomezlop@um.es; 3Campus of International Excellence “Mare Nostrum”, University of Murcia, 30720 Murcia, Spain

**Keywords:** sport, physical activity and sport in youth, executive functions, academic performance

## Abstract

It is well known that curricular physical activity benefits children’s executive functions and academic performance. Therefore, this study aimed to determine whether there is an influence of extracurricular sports on executive functions and academic performance. However, it is less known which specific types of the sport better enhance executive functions in children; to investigate this issue, this study compared the performance on executive functions tasks and academic performance in one hundred and two boys and girls with an average age of 11.84 years recruited from Italian schools and gyms (*N* = 102), who participated in martial arts or team sports or were sedentary children. Executive functions were measured with the tests: Attenzione e Concentrazione, Digit Span test, Tower of London, IOWA Gambling task BVN 5-11, and BVN 12-18. Results demonstrated that children practicing martial arts showed better executive functioning and higher school marks than those involved in team sports or not involved in any sports. Furthermore, participants aged 12 to 15 years old outperformed in cool and hot executive functions tasks and had a better academic performance. Thus, the present findings supported the view that regular practice of extracurricular sports enhances executive functions development and consequently influences academic performance.

## 1. Introduction

The World Health Organization (WHO) has identified the school as a key setting for the promotion of physical activity (PA) in children [1]. The school setting is an ideal environment where develop PA interventions in childhood [2]. PA in schools has not only advantages from a child’s physical and emotional development, but at the same time, it has also potentially wide school benefits [1,3,4,5]. Moreover, recent studies focus on the positive effect of extracurricular sport on AP [6,7]. It has been proven that sport has beneficial effects upon concentration, enhanced cognitive performance, and better academic performance (AP) [7,8,9,10].

Haapala [11] carried out a systematic literature review and analyzed studies from 1966 to 2011 on the PA, AP, and Cognition in Children and Adolescents. In summary, the author demonstrated how 14- to 36-week physical exercise training programs enhanced children’s performances in mathematical, reading, and language. Within the study’s statistical analysis, it was found that there was no direct relationship between physical activity and academic performance in the population group studied. This Relationship is supposed to be indirect and mediated by the action of the executive functions (EF) [12,13], which are considered a crucial and robust indicator of school success from preschool age [14,15,16,17,18].

EFs are a collective term under which a broad spectrum of cognitive abilities is summarized. These abilities are needed for goal-directed, purposeful thoughts and behaviors [19,20,21]. Working memory, inhibition, and cognitive flexibility are considered three core EFs and are classified as “Cool” [22] because they refer to the cognitive components of self-regulation. In addition to these, effortful control defines the “Hot” EFs of self-regulation that are involved in processing positive or negative emotional cues to achieve a goal [23,24]. EFs depend on the prefrontal cortex (PFC) area. This region’s structural growth develops slowly across development; from childhood, when it results less specialized, through adolescence with synaptic pruning processes up to early adulthood, characterized by complete maturation [25,26]. Hot and Cool EFs follow a bell-shaped curve, with an ascendant line from early adolescence and a peak in middle childhood, between 14 and 15 years old [27].

Evidence suggests that different types of sport may influence EFs in various ways: movement, motivational and emotional enhancement, social interaction, and cooperation [28,29,30]. Currently, several researchers have proposed that EFs’ improvements via sport depend on the movement characteristics involved in the activities [31,32]. Since the environment influences motor skills, sports can be classified as open or closed skills [9]. Open skills sports involve complex cognitive processes, such as perception, attention, planning, strategy development, set changes, ongoing adaptability, and active decision making [33]. They are performed in dynamic and unpredictable environments [9,34]. Basketball, tennis, karate, soccer, and volleyball squash are examples of typical open-skill exercises. Conversely, closed skills involve fewer cognitive demands [33], are performed in a predictable environment [9], and are self-paced [33]. Typical exercises are running, cycling, and swimming [9]. Recent studies suggest that open-skill sports improve cognitive functions, such as attention, perception, short-term memory, as well as EFs, as cognitive flexibility, and inhibition [9,35,36]. People involved in open-skill exercises, who are required to adapt in a constantly changing environment, are hypothesized to have better EFs than those who participate in closed-skill ones [33,37,38].

Despite the increasing occurrence of studies that demonstrate the positive effects of PA on the cognitive domain, results concerning the influence of PA on school achievement are variegated [39,40,41], although no study showed PA having detrimental effects, but at least no effects [42].

Becker and colleagues [7] found a positive association between sports, EFs, and AP, particularly in math. According to their results, participating in various open-skill sports (i.e., basketball, martial arts, or football) was linked to higher mathematic marks. The authors explained this relationship as increased spatial skills that could be connected to performance on math-related tasks and involvement in open-skill sports. They did not find a significant outcome for literacy since phoneme and letter identification of reading are processes that may less benefit from EFs. In their study, Alesi and colleagues (2014) [43] showed that participants who were involved in karate performed better on working memory, attention, and executive functions tasks.

Moreover, tennis play was associated with better working memory, cognitive flexibility, and inhibition [6]. Dyer and colleagues (2017) [44] found a positive relationship between sport and AP. Their results showed that sports participation predicted English and math performance.

Despite these results from the studies mentioned earlier, little is known about AP as influenced by specific sport categories. For example, research has supported team sports [44]; on the other hand, many studies have emphasized the benefits of martial arts [45,46,47].

In conclusion, specific sports require different demands related to varying cognitive loading [48].

The purpose of this study is to compare the performance of Cool EF and Hot EF tasks and school achievement, among participants involved in martial arts, team sports, and sedentary children with ages ranging from 7 to 15 years.

Difference among the two sports groups was hypothesized; in particular, that participants involved in martial arts would show better performance on tasks measuring EFs when compared with peers practicing team sports. Moreover, all sports groups would show better performance on EF tasks compared with sedentary peers. Lastly, it was hypothesized that better performance on EFs tasks reflects better school marks on mathematical and linguistic school subjects. As concerning age, the 12-to-15-years-old group was hypothesized to outperform on EF tasks.

## 2. Materials & Methods

### 2.1. Participants

A total of 102 Italian school children (58 boys, 44 girls) with an average chronological age of 11.84 (2.41) years were involved in this study.

Participants were classified into two age groups 7–11 (43.1%), 12–15 (56.9%). In addition, 40.2% did martial arts, 18.6% were sedentary, and 41.2% practiced team sports. As concerning gender, 56.9% were male and 43.1% were female. Inclusion criteria were the absence of intellectual disability, visual or neurological impairment, and/or neurodevelopmental disorders; and to correctly understand and speak the Italian language.

Participants practiced martial arts or team sports in their leisure time three times per week at least in the 2 years prior to the research. Sedentary children did not participate in any sport at least in the 2 years prior to the research.

The medium socio-economic level was predominant (51.7%), based on parents’ education and employment parameters. Participants were recruited in public schools or gyms, all located in an urban area, based on voluntary participation in the research project. After the headmaster of each school or gym approval, parents were contacted through flyers announcing the research project with aims, procedures, and a guarantee of anonymity and confidentiality for participants. Then parents were invited to allow their children to participate in the study and provide their written consent. This study was carried out following the recommendations of the Declaration of Helsinki (2013) and the Ethical Code of the Italian Association of Psychology (AIP).

Table 1 displays the number of participants for each age group and sport category.

### 2.2. Design

Firstly, a Socio-Demographic Schedule was administered to collect participants’ age, education, school marks, and SES. The criteria for SES evaluation were the family size, parents’ education, and job. The medium-high status was given by the parental qualification of high school or graduation and job requiring a diploma or high qualification. The medium status was given by parental qualification or middle school and high qualified job as a teacher, employee, shop assistant, or similar. The medium-low status was given from the parental qualification of primary school and qualified to require as housekeeper or student.

A battery of multiple standardized tasks was used to measure executive functioning and multiple standardized tasks were used. This consisted of the following tests: The Stroop test and The Distributed Attention test derived by the CD-ROM from Attenzione e Concentrazione [49]; The Tower of London [50]; The IOWA Gambling Task [51] derived from the Millisecond Inquisit Software; and The Digit Span and The Verbal fluency were measured using the BVN 5–11 [52] and BVN 12–18 [53].

The Digit Span Test was composed of two tasks: the Forward Digit Span, which required participants to repeat a sequence of digits in the same order of the instructions, and the Backward Digit Span, which required participants to repeat the digits in the reverse order of the instructions. Forward Digit Span consisted of 27 sequences ranging from two to nine digits, while Backward Digit Span contained 24 sequences ranging from two to eight digits. The experimenter verbally presented each sequence at the rate of one digit per second. Two trials were administered for each sequence length; if participants were correct on either trial, then they advanced to the following sequence with the number of digits increasing by one. The test was ended when participants failed on two trials of the same length. Scores were computed by counting the total number of digits successfully remembered Forward Digit Span and Backward Digit Span were analyzed as separate dependent variables, and two scores were obtained [54].

The Stroop test assessed the ability to inhibit cognitive interference of word meaning on the ability to name the color of the words appearing on the screen (blue, black, red, and green). There were two categories of stimuli fallen into two categories: congruent trials including words in which color word and the color print matched (i.e., RED painted in red color); incongruent trials including words in which color word and color print were different (i.e., RED printed in blue color). Scores resulted in reaction time and accuracy for congruent and incongruent trials.

The Tower of London (ToL) measured planning strategies. Participants were showed on the laptop screen a wooden set with three pegs and three balls (red, blue, and green), and participants were asked to move balls to reach the configuration, following specific rules (i.e., to move only one ball at a time, do not place ball outside the set, to use a predetermined set of moves). The ToL was composed of 12 trials with an increasing level of difficulty. Scores were computed by counting the correct configuration from 0 to 12 points.

The Iowa Gambling Task (IGT) is a computerized card game to assess decision-making abilities by simulating real-life situations with rewards and punishments. It is composed of four card decks: A, B, C, and D. The A and B decks are “disadvantageous,” short-term, and risky card desks. Each participant chooses a card, and s/he immediately gains money, but a high penalty follows the choice. The C and D are “advantageous”, long-term, and safe card decks. A smaller gain and lower penalty followed participant’s choice. The task is unknown to the participant, consisting of 100 trials (1 trial = 1 card drawn, 20 trials = 1 block). Long-term decision-making is reflected in the IGT score calculated as the number of cards selected from the advantageous, safe decks minus those selected from the disadvantageous, risky decks. The net score of the first 40 trials reflected uncertainty decision-making; the net score of the last 40 trials reflected risk decision-making. A high net score was given by selecting fewer cards from the disadvantageous but immediate reward decks (A and B) and drawing more cards from the advantageous reward decks (C and D). Two singles scores were obtained: 1. “good play,” given by the choice from advantageous, good, decks outweigh those from disadvantageous, bad one; 2. “bad play” given by choices from risky, bad decks exceed those from safe, good ones.

The Attenzione e Concentrazione test was used to test distributed attention through a dual-task test. Participants were instructed to push a button whenever a target appeared, simultaneously a list of words, and the child was asked to push another button when he heard the word target. Thus, attention was distributed on two parallel tasks, the visual one and the auditory one. Two scores were entered: correct answers and errors.

The Verbal Fluency Test derived consisted of category fluency and phonemic fluency to test the speed of access to the lexicon. Participants were given a minute to produce as many words as possible within a specific category (i.e., animal, fruits) or starting with a given letter (i.e., S). Scores were computed for each category (or given letter) and a total as a summary by adding all the scores together.

Tasks were administered to each participant by a researcher in a quiet room of the school/gym and required one session of 30 to 40 min. Tasks were presented in a balanced order to avoid the effect of sequence. For example, the initial order was: 1. The Digit Span; 2. The Distributed Attention, 3. The ToL; 4. The Stroop test; 5. The IOWA Gambling Task; 6. The Verbal fluency. Then, the order was 1. The Distributed Attention; 2. The ToL; 3. The Stroop test; 4. The IOWA Gambling Task; 5. The Verbal fluency; 6. The Digit Span, and so on.

School achievement was measured through the final school marks in linguistic and mathematical subjects reported by each participant at the end of the first four months of the school year. School marks ranged from 1 to 10, with higher scores indicating better school achievement. The researcher verified the matching between the self-reported school marks and school report cards.

### 2.3. Statistical Analysis

SPSS software (Released 26, SPSS Inc. NY, USA) was used to run statistical analyses. Multivariate ANOVA was computed to compare cognitive outcomes in children involved in martial arts, team sports, and sedentary children. The independent variables were sport and age, and the dependent variables were the scores on Hot EFs and Cool EFs tasks, such as working memory, inhibition, planning, attention, and decision making. The significance level was set at *p* ≤ 0.05.

## 3. Results

Data analysis showed significant differences between the three groups for the following cognitive abilities. As concerns the age main effect, 12-to-15-year-old participants had higher scores in working memory [F (1102) = 13.137, *p* = 0.000, η^2^p = 0.123], inhibition [F(1102) = 6.230, *p* = 0.014, η^2^p = 0.062], verbal fluency [F(1102) = 19.106, *p* = 0.000, η^2^p = 0.169], and in auditory distributed attention [F(1102) = 3.771, *p* = 0.05, η^2^p = 0.039]. Moreover, significant differences were found on the IOWA test in the criterion of differences between good and bad play [F(1102) = 10.074, *p* = 0.002, η^2^p = 0.097]. Results are showed in Table 2. As concerns the sport category main effect, children who practiced martial arts performed better in working memory [F(2102) = 3.680, *p* = 0.029, η^2^p = 0.073], inhibition [F(2102) = 10.891, p = 0.000, η^2^p = 0.188], distributed attention [F(2102) = 6.410, *p* = 0.002, η^2^p = 0.120], visual distributed attention [F(2102) = 6.921, *p* = 0.002, η^2^p = 0.128], and auditory distributed attention [F(2102) = 5.120, *p* = 0.008, η^2^p = 0.098]. Regarding Hot EFs, a significant difference on IOWA test in good play [F(2102) = 3.232, *p* = 0.04, η^2^p = 0.064] was found. Results are showed in Table 3.

No interaction effects of factors were found.

School marks in linguistic and mathematics were higher in children involved in martial arts than their peers involved in team sport and sedentary children. The martial arts group outperformed in linguistic, both at the ages of 7-to-11-years-old (M = 8.56) and 12-to-15-years-old (M = 7.59). In mathematics, only 7-to-11-years-old children practicing martial arts reported higher scores (M = 8.56) compared to other groups. The means and standard deviations are summarized in Table 4, separately for martial arts, team sports, and sedentary.

## 4. Discussion

This study aimed to compare performance on EFs and AP of children practicing martial arts, team sports, and sedentary children with an average age of 11.84 years. Our findings show that the 12-to-15-years-old group would perform better in EF tasks, both Cool and Hot. Higher performances were found in working memory, inhibition, verbal fluency, distributed attention, and decision-making tasks. This is coherent with developmental trajectories showing how a gradual linear maturation defines EFs from preschool throughout adolescence [55].

The gradual growth of EF is widely due to the involvement of the prefrontal cortex (PFC), particularly in the dorsolateral region. Structurally, this region develops slowly across age; from the age of six years, an activation of PFC, although less specialized, is evident when children perform EFs tasks, but this development carries on with the age until reaching the complete maturation in early adulthood age [25,26]

Moreover, participants practicing martial arts displayed a better performance on EF tasks and AP than those who attended team sports or were sedentary. In addition, they performed better in inhibition, working memory, distributed attention, verbal and auditory, verbal fluency tasks, and decision making.

Historically, martial arts have highlighted the importance of self-regulation abilities, and for this reason, the terms self-control and discipline are strongly connected to them [56]. The ability to learn self-control might be recast as the ability to inhibit, repress, control automatic responses, and create responses by using attention and reasoning. In martial arts, attention is focused on the main object, such as the rival, but at the same time distributed across its various features [56]. Moreover, martial arts and other open-skills sport are defined by changing movements and situations, stimulating goal-directed behaviors. As stated in a previous study, children practicing karate regularly show better cognitive functioning in visual selective attention, verbal working memory, reaction time; all these cognitive abilities improve karate performance and differentiate athletes and novices [57]. Specifically, successful karate performance needs high cognitive engagement and EFs components, such as updating and monitoring information in memory, switching attention resources from one task to another, and inhibiting automatic thoughts and behaviors. Moreover, karate triggers goal-directed behaviors to face changing situations and movements.

In the study, a significant benefit was observed for AP, and it consisted in higher marks on linguistic and mathematic areas for children regularly practicing extra-school sports; these results are in line with previous researches and reinforce the beneficial effects of PA on EFs and as a consequence on AP [46,47,57,58]. Moreover, children practicing martial arts had the highest school performance. Based on research, we conjecture that martial arts may improve cognitive development by different pathways: stimulating changes in the brain structure; offering an “enriched environment”; generating social interaction opportunities; enhancing self-confidence and self-awareness [59]. In addition, sports produce benefits on health and EFs and promote the enjoyment, higher self-esteem levels, and social inclusion [60,61].

To sum up, based on our results, PA as martial arts would be helpful to improve school abilities, not only as AP but also as school readiness. As proof of this, Pinto-Escalona et colleagues [47] implemented a one-year karate intervention in five different European countries. In the school gyms, an enriched environment was designed to provide senso-motor stimuli for the motor skills growth and cognitive performance, and the intervention group was trained for 2 h/week on a standard education curriculum through karate exercises. Their significant results included cognitive functioning and physical fitness and AC with a more significant increase in school marks.

### Limitations and Future Lines of Research

Interesting implications on the educational fields are supported. Firstly, coherently, it is imperative to encourage sport participation at a young age. Specific parent or teacher education programs would be necessary to share good practices deriving from the awareness of sports benefits. Too often, these benefits are linked only to physical health, as the limitation of overweight and obesity for an increasingly inactive lifestyle on childhood. Teachers and families need to understand how sports activities can contribute to child cognitive development by enhancing planning, sustained and divided attention, working memory, inhibition abilities by enjoying activities. In a complementary way, researchers and practitioners need to understand what sports and exercise interventions are suitable for cognitive development to create the perfect matching between one sports task and one executive function. In particular, as concern the martial arts, it would be remarkable to study the cognitive profiles of children practicing different components such as kumitè and kata characterized by different levels of open or closed skill activities, different goals, and different exercises training programs.

## 5. Conclusions

To sum up, the findings of this study suggest how sport participation and the practice of martial arts, in particular, contribute to improving school abilities. This study’s strength is highlighting the influence of participation in after-school sports activities on the development of EFs and on academic performance. With the recognized benefits of physical fitness for health since childhood, most studies have focused on the benefits of school physical activities.

Furthermore, future research is needed to remedy the shortcomings of this study. A significant limitation lies in methodological concerns as the measurement of children’s AP through school grades at the end of the term. Even if the grades were reliable, direct measurements of academic skills such as reading, writing, and arithmetic should be made. This is because EFs directly influence the acquisition of the school-skills as mentioned above, which, together with other contextual and social factors, ultimately result in school grades.

More future research is needed to fully understand the link between sport, EFs, and AP. Results from this study further our understanding of martial arts, which can enhance cognitive development, in particular EFs. Martial arts represent an exercise program supporting life-long physical and psychological positive effects, with significant consequences on EFs and AP.

From the psycho-educational point of view, it is essential promoting participation in sports. It has been shown to significantly increase opportunities for children to be physically active, both in curriculum physical activity and extracurricular sports [62,63].

## Figures and Tables

**Table 1 ijerph-18-11745-t001:** Number of participants for age group and sport category.

	Martial Arts	Sedentary	Team Sports	Total
**Age**				
7–11	13	6	25	44
12–15	28	13	17	58
N	41	19	42	102

**Table 2 ijerph-18-11745-t002:** Means, standard deviations, and ANOVA results for EFs tasks stratified by age.

	Age						
Executive Functions Tasks	7–11		12–15				
	M	SD	M	SD	F	*p*	η^2^p
Working memory	3.81	(1.31)	5.03	(1.21)	13.137	0.000 ***	0.123
Inhibition	28.24	(8.74)	35.00	(12.28)	6.320	0.014 **	0.062
Verbal Fluency	41.81	(9.25)	51.55	(9.18)	19.106	0.000 ***	0.169
Auditory distributed attention	4.76	(1.41)	5.34	(1.61)	3.771	0.05 *	0.039
IOWA difference between good and bad play	−2.41	(6.95)	1.90	(6.56)	10.074	0.002 **	0.097

* = *p* < 0.05; ** = *p* < 0.01; *** = *p* < 0.001.

**Table 3 ijerph-18-11745-t003:** Means, standard deviations, and ANOVA results for EFs tasks stratified by groups.

	Sport Category								
Executive Functions Tasks	Martial Arts		Team Sports		Sedentary				
	M	SD	M	SD	M	SD	F	*p*	η^2^p
Working memory	5.13	(1.41)	4.00	(1.21)	4.42	(1.26)	3.680	0.029 *	0.073
Inhibition	39.00	(14.45)	28.50	(5.00)	26.21	(6.73)	10.891	0.000 ***	0.188
Verbal Fluency	49.13	(11.73)	45.55	(8.85)	48.26	(10.30)	0.068	0.935	0.001
Distributed attention	11.08	(4.42)	8.81	(1.29)	8.32	(1.45)	6.410	0.002 **	0.120
Auditory distributed attention	5.69	(2.14)	4.83	(.58)	4.47	(1.21)	5.120	0.008 **	0.098
IOWA good play	71.89	(16.03)	79.27	(13.48)	78.83	(15.76)	3.232	0.044 *	0.064

* = *p* < 0.05; ** = *p* < 0.01; *** = *p* < 0.001.

**Table 4 ijerph-18-11745-t004:** School marks in linguistic and mathematics for the three-sport categories: martial arts, team sport, and sedentary.

		Sport Category		
School Marks	Age	Martial Arts	Team Sports	Sedentary
Linguistic	7–11 12–15	8.56 (0.73) 7.59 (1.55)	8.04 (1.10) 7.12 (0.78)	7.50 (1.38) 7.38 (1.26)
Mathematics	7–11 12–15	8.56 (0.88) 7.48 (1.28)	8.04 (1.219) 7.65 (0.93)	6.83 (1.17) 7.15 (1.28)

## Data Availability

The data presented in this study are available on request from the corresponding author. The data are not publicly available due to privacy restrictions.

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
