# Peer review of "Sports, Executive Functions and Academic Performance: A Comparison between Martial Arts, Team Sports, and Sedentary Children"

_ijerph, 2021, doi:10.3390/ijerph182211745_

Round 1
Reviewer 1 Report
Comments and suggestions are included in the attached document.
Best regards

Author Response
Thank you very much for offering us the opportunity to resubmit the new version of our manuscript. We would like to offer our sincerest gratitude to the Reviewers for constructive suggestions that we used to make revisions in response to the raised concerns. We hope that the manuscript is now suitable to be published on International Journal of Environmental Research and PublicHealth.
Following changes have been made and written in red
As suggested line numbers have been added.
REVIEWER 1
1. An author for correspondence is not identified, nor is the address to which correspondence should be addressed.
- author for correspondence has been added (line 8)
ABSTRACT
- The authors have provided an abstract in accordance with the requirements, however, I strongly recommend that the authors strive to present an abstract with a clearer thread and highlight the objective of the study. I also recommend detailing the tests performed to measure executive functioning.
- This section was modified and a detailed list of the test used was added (lines 11-25)
- In the last sentence it is stated that "there are other ways in which physical activity influences academic performance through executive functions", please try to provide what "other ways" you are referring to.
- This sentence was modified (lines 23-25)
INTRODUCTION
- The introduction of the manuscript is very long and could be more concise. Some paragraphs lack relevant information about study object, since they refer to different population groups, with specific training programs during controlled time periods or practitioners of different sports than those taken as reference in the sample, and, therefore, not necessarily comparable with the study population.
- The introduction has been modified and summarized.
- In this section there is also no clear thread that addresses the background in relation to the topic studied and between the different ideas presented and addressed. It also includes irrelevant and sometimes unspecific information on the topic under study.
- As you suggested, all irrelevant and unspecific information has been deleted.
- It is recommended that the authors review in the second paragraph of this section when they say that "The relationship between PA and PA is supposed to be indirect and is mediated by..." and make sure if that relationship is direct or indirect; we suggest changing the wording for greater clarity to "within the statistical analysis of the study it was found that there is no direct relationship between physical activity and academic performance in the population group studied".
- As suggested, this sentence has been added (lines 44-46)
MATERIALS AND METHODS
- The methods section contains the necessary information on the tests and trials that were carried out, describes what they consist of and the attempts made, as well as which of them were selected. Although, due to its length, it is recommended to make structuring by sections (in addition to the already included 2.1 Participants") such as design, statistical analysis, etc., but in general it is not very concise and some essential aspects are not clear, compromising one of the essential principles of the scientific method, such as the principle of reproducibility, which refers to the ability to reproduce or replicate an experiment or research.
- Three sections were added and the order was changed: 2.1 participants (line 106); 2.2 design (line 130); 2.3 statistical analysis (line 202)
- The two paragraphs preceding item 2.1 are examples of this lack of conciseness "the above tasks were administered individually" without specifying whether the person administering them was a single person or whether there were more interviewers, "the tasks were presented in a balanced order to avoid the effect of the sequence", it is recommended to indicate what that order was or that the grades in the subjects were provided by the same subjects of the study without any verification of their veracity.
- In the two paragraphs (in 2.2), before preceding the 2.1, has been added the information requested (line 191). Also the order of test presentation (lines 193-196) and the verification of school marks has been specified (194-195). The description of tests was summarized and clarified to respect the principle of reproducibility.
- As for the participants, it is not specified how the sample of 102 schoolchildren was selected, what population they represented, whether they belonged to public or private schools, or to rural or urban localities.
- The description of participants selection has been specified (lines 118-120). Among the inclusion criteria we deleted “caucasian race” and we replaced with “to understand and speak the Italian language correctly”. This sentence best describes the basis of the inclusion criteria we adopted (lines 112-113).
- The expression "the middle socioeconomic level predominated" is used without the authors providing information on how these socioeconomic levels are structured. It is recommended that the term "predominated" be changed because it is not very specific in terms of the % of that level.
- Information about how socio-economic levels were structured and percentage has been added (lines 131-137). The percentage has been added (line 117).
- Furthermore, although it is indicated that the research was carried out in accordance with the guidelines of the Helsinki declaration, there is no reference to the research having been approved by an ethics committee, nor is the registration number indicated if it was carried out.
- As declared in the submission phase of the manuscript, when the research was carried out at the University of Palermo The Ethical Committee (Comitato di Bioetica dell’Università degli Studi di Palermo https://www.unipa.it/strutture/comitato-bioetica/) was missing. It started in April 2020 (Decreti Rettorali n. 1044 del 01 aprile 2020 e n. 1127 del 16 aprile 2020). The data of this research were collected before this date, so we were unable to have an ethical approval code. Anyway, as we declared in the manuscript, we respected the recommendations of the Declaration of Helsinki (2013) and the Ethical Code of the Italian Association of Psychology (AIP http://www.aipass.org/node/11560).
RESULTS
- The first paragraph I suggest taking it as a "statistical analysis" section to the "Methods" section.
- As you suggested, the first paragraph has been transferred in the material and method section and it was recalled “2.3 Statistical Analysis” (lines 202-207)
- In the second paragraph "In addition, significant differences were found in the IOWA........ test. The results are shown in Table 1." This reference is not correct. I suggest changing it to the corresponding table (table 2).
- Further on, "as for hot FEs, a significant difference was found in the IOWA test in the good game....... The results are shown in table 2". I suggest changing to the corresponding table (table 3).
- In the last paragraph of the section, in its last lines "Means and standard deviations are summarized in table 3, separately for martial arts, team sports and sedentary". I suggest that this be changed to the corresponding table (table 4).
2.3.4. All the table references have been corrected.
- Most of the statistical data provided in this section do not appear in the tables, so it can be considered that the tables do not appear complete and, therefore, makes their evaluation difficult.
- Table 2 and 3 have been modified adding the required information. All the results were showed in the new tables (lines 223 and 227).
DISCUSSION
- The title of the section states "4. CONCLUSION". Undoubtedly this is an error, I suggest changing it to "4. DISCUSSION".
- Sorry for this oversight, the right title is discussion not conclusion. It has been corrected (line 242)
- Even more so, considering that in the design phase there are some important shortcomings that were not considered and that could alter or explain the results and that I highlight in the following questions: could it be that the subjects who perform the different physical activities have different socioeconomic levels due to the price of performing one or the other? could it be that these socioeconomic differences, which the authors attribute to the benefits of karate, determine educational differences that mark the differences in executive functions and/or academic performance? Would they have any influence on the results if they come from different environments (public/private schools, rural/urban city of residence)? Could the cognitive differences noted between the 7-11 and 12-15 age groups be simply the result of the chronological developmental trajectory or could there be, according to the authors, some other cause derived from the object of study?
2.We referred to gradual growth (lines 249-253). Other modification have been done in all the section.
- I suggest using the conditional verbs "could show", "would perform better", "would justify" "could be useful".
3.As suggested, conditional verbs has been used.
- Consider deleting the ratification of your results by relying on studies of non-homogeneous populations "adults", affected by "autism spectrum disorders" or "ADHD population".
4.This sentence has been deleted
- I also recommend that the authors add a paragraph at the end of this section indicating the main limitations of the study and future lines of research.
5.A new paragraph containing limitation and future lines of research has been added (line 289-302)
CONCLUSIONS
- The conclusion section should be clear and concise and should respond to the proposed objective. I consider that clarity is compromised by citing open ability sports and specifically highlighting karate, when the article (introduction section) also cites team sports as open ability, without providing any opinion on what the difference between the two might be due to. Also, the possible implications of the results obtained make more sense in the Discussion section. Please consider this and modify this section accordingly.
- This section has been modified (lines 305-323)
REFERENCES
- The references used throughout the manuscript are sufficient for the main topic addressed. However, I recommend that authors consult the journal guidelines to modify the style of the references in accordance with these guidelines. Thus, for example, the following should be modified: punctuation marks between authors in reference number 60. In addition, the names of some journals are written as full titles and others as abbreviations. Please modify it, use ISO4 to abbreviate journal titles. The DOI (Digital Object Identifier) does not appear in any reference
- References has been modified following the guidelines.
Reviewer 2 Report
This study looked to compare the executive function and academic performance of school aged children who take part / do not take part in sport. I enjoyed reading the manuscript and thank you for preparing it for publication. I have made a number of comments below for your consideration.
Abstract
Many of my below comments have implications for the abstract and therefore this will probably need to be rewritten.
Introduction
In the introduction, and throughout the manuscript, ‘physical activity’ (or ‘PA’), ‘exercise’, ‘sport’ and ‘sedentary’ are used interchangeably. It seems to me that from the title and the research presented in this manuscript, the emphasis is on the relationship between sport, EF and AP. Therefore, can the introduction be written to focus on current literature exploring the relationship between sport (not physical activity and exercise) and EF and AP in young people.
The relevance of the school is not clear given that you are writing about sport and not physical education. I do not think the current study is about school-based physical activity. The Introduction would instead benefit from a paragraph which presents published literature on the increasing focus on academic performance in school, and the need to justify the need for PA in youth (be that physical education or participation in sport) as opposed to taking part in sport for intrinsic reasons. Has academic achievement testing reduced opportunities for children to be physically active during and after school?
Can the introduction highlight why the particular age range 7-15 years is a focus in this current study (using existing literature to support this), and why is it hypothesised that 12-15 years will perform higher on EF and how does this relate to participation in sport?
Towards the end of the introduction, a series of short paragraphs are presented (some of which are only one or two sentences in length). Please look to restructure / rewrite this section so that a more coherent argument is presented.
Materials and methods
The Tower of London reference is confusing.
Participants were categorised into three groups. For the martial arts and team sport groups, did these children take part in these sport in school or out of school? It is also unclear how it the sedentary group was established – were these children who do meet the current PA guidelines or were they children who do not take part in sport?
Furthermore, at what level and how often did the children categorised as those who participate in martial arts and team sports take part in their sports? If you did not collect these data then perhaps this is a limitation of the research and our ability to infer from findings (young athletes can operate at very different levels of participation and performance within sport, which would logically impact EF)
Why did being Caucasian form part of the inclusion criteria. On what scientific basis was this decision made?
Results
This is related to one of my earlier points regarding the Introduction, but it is unclear why older and young children were compared using inferential statistics. The meaningful comparison for the purposes of this study is whether there is an age difference for EF according to sport participation. Could Tables 2 and 3 be adapted to enable comparison between sport participation (as in Table 4).
Discussion
In the paragraph written ‘Our study results are coherent with previous studies involving adult participants [46,57] and also clinical population, in autism spectrum disorder [62,63,64] and ADHD population [65]’, please explain why there is coherence.
The Discussion seems to me to miss the point that young people take part in sport for a variety of reasons, not just because of their health and cognitive performance. Therefore, I suggest that the following paragraph is re-written in order to present a case for schools to provide structured activity to benefit EF and AP: ‘From the psycho-educational point of view, it's important promoting participation in PA. It's crucial for researchers to understand which type of PA interventions lead to improvements in children, both from a physical and cognitive perspective’.
The Discussion will need to be rewritten/edited in line with my earlier comments on the introduction, methods and results.
Grammar and syntax
The manuscript would benefit from a through proofread as some sentences do not make complete sense to the reader. For example, the 5th sentence in the Abstract (starting ‘for the purpose of investigate…) does not make sense and the sentence at the end of the opening paragraph of the Introduction may be better expressed as: ‘In particular, it has been proven that PA has beneficial effects upon concentration, enhanced cognitive performance, and better academic performance (AP) [6,7,8,9].’ Also, for example, the last sentence of the second paragraph in the Materials and methods section does not scan – I suggest splitting this sentence into two sentences at the semicolon, removing ‘likewise’ and completing the last part of the sentence in order to clarity what the Forward Digital Span test allows children to do.
Some sentences have words missing (e.g., ‘years’ missing when stating the age of participants). And ‘researches’ was used twice in the manuscript, but I think either ‘research’ or ‘researcher’ is meant.
Author Response
Thank you very much for offering us the opportunity to resubmit the new version of our manuscript. We would like to offer our sincerest gratitude to the Reviewers for constructive suggestions that we used to make revisions in response to the raised concerns. We hope that the manuscript is now suitable to be published on International Journal of Environmental Research and PublicHealth.
Following changes have been made and written in red
As suggested line numbers have been added.
REVIEWER2
Abstract
- Many of my below comments have implications for the abstract and therefore this will probably need to be rewritten. 1.This section has been rewritten (lines 11-25)
Introduction
- In the introduction, and throughout the manuscript, ‘physical activity’ (or ‘PA’), ‘exercise’, ‘sport’ and ‘sedentary’ are used interchangeably. It seems to me that from the title and the research presented in this manuscript, the emphasis is on the relationship between sport, EF and AP. Therefore, can the introduction be written to focus on current literature exploring the relationship between sport (not physical activity and exercise) and EF and AP in young people. 1.As suggested, this section has been rewritten focusing on the current literature on sport, EF and AP (lines 38, 77-87)
- The relevance of the school is not clear given that you are writing about sport and not physical education. I do not think the current study is about school-based physical activity. The Introduction would instead benefit from a paragraph which presents published literature on the increasing focus on academic performance in school, and the need to justify the need for PA in youth (be that physical education or participation in sport) as opposed to taking part in sport for intrinsic reasons. Has academic achievement testing reduced opportunities for children to be physically active during and after school?
2.Information required has been added
- Can the introduction highlight why the particular age range 7-15 years is a focus in this current study (using existing literature to support this), and why is it hypothesised that 12-15 years will perform higher on EF and how does this relate to participation in sport? 3. The reason why we selected this age range has been explained in the text (55-59)
- Towards the end of the introduction, a series of short paragraphs are presented (some of which are only one or two sentences in length). Please look to restructure / rewrite this section so that a more coherent argument is presented. 4.This section has been modified.
MATERIALS AND METHODS
- The Tower of London reference is confusing.
1.The description of the tests used has been modified.
- Participants were categorised into three groups. For the martial arts and team sport groups, did these children take part in these sport in school or out of school? It is also unclear how it the sedentary group was established – were these children who do meet the current PA guidelines or were they children who do not take part in sport? Furthermore, at what level and how often did the children categorised as those who participate in martial arts and team sports take part in their sports? If you did not collect these data then perhaps this is a limitation of the research and our ability to infer from findings (young athletes can operate at very different levels of participation and performance within sport, which would logically impact EF)
2.Information has been added (lines 114-116)
- Why did being Caucasian form part of the inclusion criteria. On what scientific basis was this decision made?
3.It has been modified (lines 112-113)
RESULTS
- This is related to one of my earlier points regarding the Introduction, but it is unclear why older and young children were compared using inferential statistics. The meaningful comparison for the purposes of this study is whether there is an age difference for EF according to sport participation. Could Tables 2 and 3 be adapted to enable comparison between sport participation (as in Table 4). 1. Information has been added. Table 2 and 3 has been modified (lines 223 and 227)
DISCUSSION
- In the paragraph written ‘Our study results are coherent with previous studies involving adult participants [46,57] and also clinical population, in autism spectrum disorder [62,63,64] and ADHD population [65]’, please explain why there is coherence. 1.This section has been deleted as suggested by the Reviewer 1
- The Discussion seems to me to miss the point that young people take part in sport for a variety of reasons, not just because of their health and cognitive performance. Therefore, I suggest that the following paragraph is re-written in order to present a case for schools to provide structured activity to benefit EF and AP: ‘From the psycho-educational point of view, it's important promoting participation in PA. It's crucial for researchers to understand which type of PA interventions lead to improvements in children, both from a physical and cognitive perspective’.
- This paragraph has been re-written and enriched with the information required (lines 278-280). The sentence “From the psycho-educational point of view, it's important promoting participation in PA. It's crucial for researchers to understand which type of PA interventions lead to improvements in children, both from a physical and cognitive perspective” has been modified and moved to conclusion section (lines 320-323).
- The Discussion will need to be rewritten/edited in line with my earlier comments on the introduction, methods and results. 3.This section has been rewritten as you suggested.
Round 2
Reviewer 1 Report
COMMENTS
Thanks again to the authors for their submission to IJERPH and at this very moment for the interest shown in trying to respond to the suggestions and proposals made by the reviewers. This new version of the article on the interrelationship between executive functions (EF), physical activity (PA) -karate, team sports and sedentary lifestyle- and academic performance (AP) in Italian children between 7 and 15 years of age, I consider that it has gained in clarity and coherence, making the result more interesting and easier to follow and interpret for the readers of the journal.
I fully acknowledge the effort involved in adapting the wording to the comments that were made. I understand that this work has been satisfactorily completed, sometimes by accepting the reviewer's recommendations and other times with the authors' own contributions, which undoubtedly improve the result.
SPECIFIC COMMENTS
Most of the comments, suggestions and recommendations made have been addressed and the manuscript is now satisfactorily considered for publication, with two minor modifications to be made, which will only take a few minutes (see below):
1. Lines 263-264: Adapt the bibliographical references to the format required by the journal, unifying it with the rest of the article. Consider modifying "(25 Harms et al., 2014; 26Boelema et al., 2014)." to "[25-26].".
2. Lines 340-341: Adapt the bibliographical references to the format required in the journal, unifying it with the rest of the article. Consider modifying "(62, 63 Quick et al., 2010; The Loughborough Partnership, 2008)" to "[62-63].". Be sure to close the sentence with a full stop.
Kind regards,

Author Response
Thank you very much for the opportunity to improve our article according your suggestion. All bibliographical references has been modified.

Reviewer 2 Report
Thank you for considering and responding to my previous comments. I hope you agree that this manuscript is now improved and more focused.
There are still a few errors with grammar and syntax which an editorial team can address with you or on your behalf. Also, there still remains a series of short (1-2 sentence) paragraphs at the end of the Introduction which need attention to improve the flow and the argument being presented.
Author Response
Thank you very much for your comments and suggestions.
Short paragraphs at the end of the introduction have been deleted. Editing of English language has been done
